# Guidance on how to develop complex interventions to improve health and healthcare

Alicia O'Cathain,[1] Liz Croot,[1] Edward Duncan,[2] Nikki Rousseau,[2] Katie Sworn,[1] Katrina M Turner,[3] Lucy Yardley,[3,4] Pat Hoddinott[2]

[1]Medical Care Research Unit, School of Health and Related Research, University of Sheffield, Sheffield, UK
[2]Nursing, Midwifery and Allied Health Professional Research Unit, University of Stirling, Stirling, UK
[3]School of Social and Community Medicine, University of Bristol, Bristol, UK
[4]Psychology, University of Southampton, Southampton, UK

**Correspondence to**
Professor Alicia O'Cathain;
a.ocathain@sheffield.ac.uk

## ABSTRACT

**Objective** To provide researchers with guidance on actions to take during intervention development.
**Summary of key points** Based on a consensus exercise informed by reviews and qualitative interviews, we present key principles and actions for consideration when developing interventions to improve health. These include seeing intervention development as a dynamic iterative process, involving stakeholders, reviewing published research evidence, drawing on existing theories, articulating programme theory, undertaking primary data collection, understanding context, paying attention to future implementation in the real world and designing and refining an intervention using iterative cycles of development with stakeholder input throughout.
**Conclusion** Researchers should consider each action by addressing its relevance to a specific intervention in a specific context, both at the start and throughout the development process.

## INTRODUCTION

There is increasing demand for new interventions as policymakers and clinicians grapple with complex challenges, such as integration of health and social care, risk associated with lifestyle behaviours, multimorbidity and the use of e-health technology. Complex interventions are often required to address these challenges. Complex interventions can have a number of interacting components, require new behaviours by those delivering or receiving the intervention or have a variety of outcomes.[1] An example is a multicomponent intervention to help people stand more at work, including a height adjustable workstation, posters and coaching sessions.[2] Careful development of complex interventions is necessary so that new interventions have a better chance of being effective when evaluated and being adopted widely in the real world. Researchers, the public, patients, industry, charities, care providers including clinicians and policymakers can all be involved in the development of new interventions to improve health, and all have an interest in how best to do this.

The UK Medical Research Council (MRC) published influential guidance on developing and evaluating complex interventions, presenting a framework of four phases: development, feasibility/piloting, evaluation and implementation.[1] The development phase is what happens between the idea for an intervention and formal pilot testing in the next phase.[3] This phase was only briefly outlined in the original MRC guidance and requires extension to offer more help to researchers wanting to develop complex interventions. Bleijenberg and colleagues[4] brought together learning from a range of guides/published approaches to intervention development to enrich the MRC framework.[4] There are also multiple sources of guidance to intervention development, embodied in books and journal articles about different approaches to intervention development (for example[5]) and overviews of the different approaches.[6] These approaches may offer conflicting advice, and it is timely to gain consensus on key aspects of intervention development to help researchers to focus on this endeavour. Here, we present guidance on intervention development based on a consensus study which we describe below. We present this guidance as an accessible communication article on how to do intervention development, which is aimed at readers who are developers, including those new to the endeavour. We do not present it as a 'research article' with methods and findings to maximise its use as guidance. Lengthy detail and a long list of references are not provided so that the guidance is focused and user friendly. In addition, the key actions of intervention development are summarised in a single table so that funding panel members and developers can use this as a quick reference point of issues to consider when developing health interventions.

## HOW THIS GUIDANCE WAS DEVELOPED

This guidance is based on a study funded by the MRC and the National Institute for Health

Research in the UK, with triangulation of evidence from three sources. First, we undertook a review of published approaches to intervention development that offer developers guidance on specific ways to develop interventions[6] and a review of primary research reporting intervention development. The next two phases involved developers and wider stakeholders. Developers were people who had written articles or books detailing different approaches to developing interventions and people who had developed interventions. Wider stakeholders were people involved in the wider intervention development endeavour in terms of being directors of research funding panels, editors of journals that had published intervention development studies, people who had been public and patient involvement members of studies involving intervention development and people working in health service implementation. We carried out qualitative interviews[7] and then we conducted a consensus exercise consisting of two simultaneous and identical e-Delphi studies distributed to intervention developers and wider stakeholders, respectively, and followed this with a consensus workshop. We generated items for the e-Delphi studies based on our earlier reviews and analysis of interview data and asked participants to rate 85 items on a five-point scale from 'very' to 'not important' using the question 'when developing complex interventions to improve health, how important is it to'. The distribution of answers to each item is displayed in Appendix 1, and e-Delphi participants are described in Appendix 2. In addition to these research methods, we convened an international expert panel with members from the UK, USA and Europe early in the project to guide the research. Members of this expert panel participated in the e-Delphi studies and consensus workshop alongside other participants.

## FRAMEWORK FOR INTERVENTION DEVELOPMENT
We base this guidance on expert opinion because there is a research evidence gap about which actions are needed in intervention development to produce successful health interventions. Systematic reviews have been undertaken to determine whether following a specific published approach, or undertaking a specific action, results in effective interventions. Unfortunately, this evidence base is sparse in the field of health, largely due to the difficulty of empirically addressing this question.[8 9] Evidence tends to focus on the use of existing theory within intervention development—for example, the theory of Diffusion of Innovation or theories on behaviour change—and a review of reviews shows that interventions developed with existing theory do not result in more effective interventions than those not using existing theory.[10] The authors of this latter review highlight problems with the evidence base rather than dismiss the possibility that existing theory could help produce successful interventions.

Key principles and actions of intervention development are summarised below. More detailed guidance for the

principles and actions is available at https://www.sheffield.ac.uk/scharr/sections/hsr/mcru/indexstudy.

### Key principles of intervention development
Key principles of intervention development are that it is dynamic, iterative, creative, open to change and forward looking to future evaluation and implementation. Developers are likely to move backwards and forwards dynamically between overlapping actions within intervention development, such as reviewing evidence, drawing on existing theory and working with stakeholders. There will also be iterative cycles of developing a version of the intervention: getting feedback from stakeholders to identify problems, implementing potential solutions, assessing their acceptability and starting the cycle again until assessment of later iterations of the intervention produces few changes. These cycles will involve using quantitative and qualitative research methods to measure processes and intermediate outcomes, and assess the acceptability, feasibility, desirability and potential unintended harms of the intervention.

Developers may start the intervention development with strong beliefs about the need for the intervention, its content or format or how it should be delivered. They may also believe that it is possible to develop an intervention with a good chance of being effective or that it can only do good not harm. Being open to alternative possibilities throughout the development process may lead to abandoning the endeavour or taking steps back as well as forward. The rationale for being open to change is that this may reduce the possibility of developing an intervention that fails during future evaluation or is never implemented in practice. Developers may also benefit from looking forward to how the intervention will be evaluated so they can make plans for this and identify learning and key uncertainties to be addressed in future evaluation.

### Key actions of intervention development
Key actions for developers to consider are summarised in table 1 and explored in more detail throughout the rest of the paper. It may not be possible or desirable for developers to address all these actions during their development process, and indeed some may not be relevant to every problem or context. The recommendation made here is that developers 'consider the relevance and importance of these actions to their situation both at the start of, and throughout, the development process'.

These key actions are set out in table 1 in what appears to be a sequence. However, in practice, these actions are addressed in a dynamic way. That is, undertaken in parallel and revisited regularly as the intervention evolves, or they interact with each other when learning from one action influences plans for other actions. These actions are explored in more detail below and presented in a logic model for intervention development (figure 1). A logic model is a diagram of how an intervention is proposed to work, showing mechanisms by which an intervention influences the proposed outcomes.[11] The short and

**Table 1** Framework of actions for intervention development

| Action | Consider the relevance and importance of the following |
| --- | --- |
| Plan the development process | Identify the problem to be targeted and refine understanding of it throughout the process.<br>Assess whether the problem is a priority.<br>Consider which aspects of the problem are amenable to change.<br>Ask whether a new intervention is really needed and if the potential benefit of the new intervention justifies the cost of development.<br>Determine the time needed to undertake intervention development.<br>Obtain sufficient resources/funding for the intervention development study.<br>Draw on one or more of the many published intervention development approaches, recognising that there is no evidence about which approach is best and apply flexibly depending on the problem and context.<br>Involve stakeholders during the planning process (see next Action).<br>Produce a protocol detailing the processes to be undertaken to develop the intervention. |
| Involve stakeholders, including those who will deliver, use and benefit from the intervention | Work closely with relevant stakeholders throughout the development process: patients, the public, the target population, service providers, those who pay for health and social services or interventions, policymakers and intervention design specialists.<br>Develop a plan at the start of the process to integrate public and patient involvement into the intervention development process.<br>Identify the best ways of working with each type of stakeholder, from consultation through to coproduction, acknowledging that different ways may be relevant for different stakeholders at different times.<br>Use creative activities within team meetings to work with stakeholders to understand the problem and generate ideas for the intervention. |
| Bring together a team and establish decision-making processes | Include within the development team individuals with relevant expertise: in the problem to be addressed by the intervention including those with personal experience of the problem, in behaviour change when the intervention aims to change behaviour, in maximising engagement of stakeholders and with a strong track record in designing complex interventions.<br>It may be hard to make final decisions about the content, format and delivery of the intervention, so only some team members may do this. There is no consensus about the size or constituency of the team that makes these final decisions, but it is important early on to agree a process for making decisions within the team. |
| Review published research evidence | Review published research evidence before starting to develop the intervention and throughout the development process for example, to identify existing interventions, to understand the evidence base for each proposed substantive intervention component.<br>Look for, and take into account, evidence that the proposed intervention may not work in the way intended. |
| Draw on existing theories | Identify an existing theory or framework of theories to inform the intervention at the start of the process, for example, behaviour change or implementation theory.<br>Where relevant, draw on more than one existing theory or framework of theories for example, both psychological and organisational theories. |
| Articulate programme theory | Develop a programme theory. The programme theory may draw on existing theories. Aspects of the programme theory can be represented by a logic model or set of models.<br>Test and refine the programme theory throughout the development process. |
| Undertake primary data collection | Use a wide range of research methods throughout, for example, qualitative research to understand the context in which the intervention will operate, quantitative methods to measure change in intermediate outcomes. |
| Understand context | Understand the context in which the intervention will be implemented. Context may include population and individuals; physical location or geographical setting; social, economic, cultural and political influences and factors affecting implementation, for example, organisation, funding and policy. |
| Pay attention to future implementation of the intervention in the real world | From the start, understand facilitators and barriers to reaching the relevant population, future use of the intervention, 'scale up' and sustainability in real world contexts. |
| Design and refine the intervention | Generate ideas about content, format and delivery with stakeholders.<br>Once an early version or prototype of the intervention is available, refine or optimise it using a series of iterations. Each iteration includes an assessment of how acceptable, feasible and engaging the intervention is, including potential harms and unintended consequences, resulting in refinements to the intervention. Repeat the process until uncertainties are resolved.<br>Check that the proposed mechanisms of action are supported by early testing. |

Continued

| Table 1 | Continued |
|---|---|
| **Action** | **Consider the relevance and importance of the following** |
| End the development phase | There are no established criteria for stopping the intensive development phase and moving on to the feasibility/pilot or evaluation phases. The concepts of data saturation and information power may be useful when assessment of later iterations of the intervention produces few changes. <br> Describe the intervention to facilitate transferability of an intervention outside the original team and location in which it was developed. <br> Write up the intervention development process so that judgements can be made about the quality of the process, links can be made in the future between intervention development processes and the subsequent success of interventions, and others can learn how it can be done |

long-term effects of successful intervention development were informed by the qualitative interviews with developers and wider stakeholders.[7]

### Plan the development process
#### Understand the problem
Developers usually start with a problem they want to solve. They may also have some initial ideas about the content, format or delivery of the proposed intervention. The knowledge about the problem and the possibilities for an intervention may be based on: personal experiences of the problem (patients, carers or members of the public); their work (practitioners, policymakers, researchers); published research or theory or discussions with stakeholders. These early ideas about the intervention may be refined and indeed challenged throughout the intervention development process. For example, understanding the problem, priorities for addressing it and the aspects

that are amenable to change is part of the development process, and different solutions may emerge as understanding increases. In addition, developers may find that it is not necessary to develop a new intervention because effective or cost-effective ones already exist. It may not be worth developing a new intervention because the potential cost is likely to outweigh the potential benefits or its limited reach could increase health inequalities, or the current context may not be conducive to using it. Health economists may contribute to this debate.

#### Identify resources—time and funding
Once a decision has been made that a new intervention is necessary, and has the potential to be worthwhile, developers can consider the resources available to them. Spending too little time developing an intervention may result in a flawed intervention that is later found not to be effective or cost-effective or is not implemented

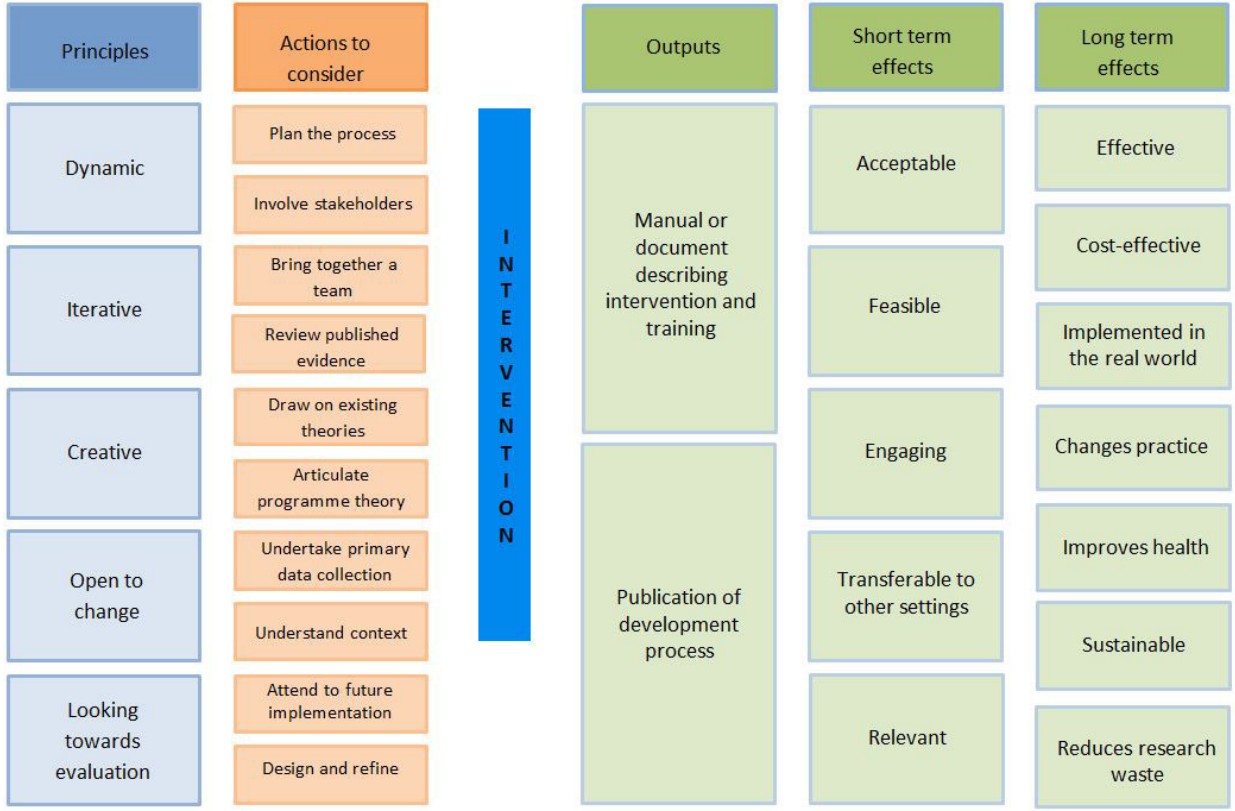

**Figure 1** Logic model for intervention development.

in practice, resulting in research waste. Alternatively, spending too much time on development could also waste resources by leaving developers with an outdated intervention that is no longer acceptable or feasible to deliver because the context has changed so much or is no longer a priority. It is likely that a highly complex problem with a history of failed interventions will warrant more time for careful development.

Some funding bodies fund standalone intervention development studies or fund this endeavour as part of a programme of development, piloting and evaluation of an intervention. While pursuing such funding may be desirable to ensure sufficient resource, in practice some developers may not be able to access this funding and may have to fund different parts of the development process from separate pots of money over a number of years.

Applying for funding requires writing a protocol for a study. Funders need detail about the proposed intervention and the development process to make a funding decision. It may feel difficult to specify the intervention and the detail of its development before starting because these will depend on learning occurring throughout the development process. Developers can address this by describing in detail their best guess of the intervention and their planned development process, recognising that both are likely to change in practice. Even if funding is not sought, it may be a good idea to produce a protocol detailing the processes to be undertaken to develop the intervention so that sufficient resources can be identified.

### Decide which approach to intervention development to take

A key decision for teams is whether to be guided by one of the many published approaches to intervention development or undertake a more pragmatic self-selected set of actions. A published approach is a guide to the process and methods of intervention development set out in a book, website or journal article. The rationale for using a published approach is that it sets out systematic processes that other developers have found useful. Some published approaches and approaches that developers have used in practice are listed in table 2.[6] No research has shown that one of these approaches is better than another or that their use always leads to the development of successful interventions. In practice, developers may select a specific published approach because of the purpose of their intervention development, for example, aiming to change behaviour might lead to the use of the Behaviour Change Wheel or Intervention Mapping, in conjunction with the Person Based Approach. Alternatively, selection may depend on developers' beliefs or values, for example, partnership approaches such as coproduction may be selected because developers believe that users will find the resultant interventions more acceptable and feasible, or they may value inclusive work practices in their own right. Although developers may follow a published approach closely, experts recommend that developers apply these approaches flexibly to fit their specific context. Many of these approaches share the same actions[4 6] and simply place more emphasis on one or a subset of actions. Researchers sometimes combine the use of different approaches in practice to gain the strengths of two approaches, as in the 'Combination' category of table 2.

### Involve stakeholders throughout the development process

Many groups of people are likely to have a stake in the proposed intervention: the intervention may be aimed at patients or the public, or they may be expected to use the intervention; practitioners may deliver the intervention in a range of settings, for example, hospitals, primary care, community care, social care, schools, communities, voluntary/third sector organisations and users, policy makers or tax payers may pay for the intervention. The rationale for involving relevant stakeholders from the start, and indeed working closely with them throughout, is that they can help to identify priorities, understand the problem and help find solutions that may make a difference to future implementation in the real world.

There are many ways of working with stakeholders and different ways may be relevant for different stakeholders at different times during the development process. Consultation may sometimes be appropriate, where a one-off meeting with a set of stakeholders helps developers to understand the context of the problem or the context in which the intervention would operate. Alternatively, the intervention may be designed closely with stakeholders using a coproduction process, where stakeholders and developers generate ideas about potential interventions and make decisions together throughout the development process about its content, format, style and delivery.[12] This could involve a series of workshops and meetings to build relationships over time to facilitate understanding of the problem and generation of ideas for the new intervention. Coproduction rather than consultation is likely to be important when buy-in is needed from a set of stakeholders to facilitate the feasibility, acceptability and engagement with the intervention or the health problem or context is particularly complex. Coproduction involves stakeholders in this decision-making, whereas with consultation, decisions are made by the research team. Stakeholders' views may also be obtained through qualitative interviews, surveys and stakeholder workshops, with methods tailored to the needs of each stakeholder. Innovative activities can be used to help engage stakeholders, for example: creative sessions facilitated by a design specialist might involve imagining what versions of the new intervention might look like if designed by various well-known global manufacturers or creating a patient persona to help people think through the experiences of receiving an intervention. As well as participating in developing the intervention, stakeholders can help to shape the intervention development process itself. Members of the public, patients and service users are key stakeholders, and experts recommend planning to integrate their involvement into the intervention development process from the start.

**Table 2** Different approaches to intervention development

| Category | Definition | Examples of approaches* |
|---|---|---|
| 1. Partnership | The people whom the intervention aims to help are involved in decision-making about the intervention throughout the development process, having at least equal decision-making powers with members of the research team. | Coproduction, cocreation, codesign; user driven; experience-based codesign; community-based participatory research |
| 2. Target population centred | Interventions are based on the views and actions of the people who will use the intervention. | Person based; user centred; human-centred design |
| 3. Theory and evidence based | Interventions are based on combining published research evidence and existing theories for example, psychological or organisational theories. | Medical Research Council Framework for developing and evaluating complex interventions; Behaviour Change Wheel; Intervention Mapping; Normalisation Process Theory; Theoretical Domains Framework |
| 4. Implementation based | Interventions are developed with attention to ensuring the intervention will be used in the real world if found to be effective at the evaluation phase. | Reach, Effectiveness, Adoption, Implementation, Maintenance |
| 5. Efficiency based | Components of an intervention are tested using experimental designs to determine active components and make interventions more efficient. | Multiphase Optimization Strategy |
| 6. Stepped or phased | Interventions are developed through emphasis on a systematic and sequential set of processes involved in intervention development. | Six essential Steps for Quality Intervention Development; Five actions model; Obesity Related Behavioral Intervention Trials |
| 7. Intervention specific | An intervention development approach is constructed for a specific type of intervention. | Digital (eg, Integrate, Design, Assess and Share); patient decision support aids |
| 8. Combination | Published approaches to intervention development are combined. | Participatory Action Research based on theories of Behaviour Change and Persuasive Technology |
| 9. Pragmatic | Developers use a self-selected set of actions. | Sometimes framed as mixed methods or formative evaluation |

*See reference 6 for references and examples.

### Bring together a team and establish decision-making processes

Developers may choose to work within any size of team. Small teams can reach out to stakeholders at different points in the development process. Alternatively, large teams may include all the necessary expertise. Experts recommend including: experts in the problem to be addressed by the intervention; individuals with a strong track record in developing complex interventions; a behaviour change scientist when the intervention aims to change behaviour and people who are skilled at maximising engagement of stakeholders. Other possible team members include experts in evaluation methods and economics. Within a coproduction approach to development, key stakeholders participate as equal partners with researchers. Large teams can generate ideas and ensure all the relevant skills are available but may also increase the risk of conflicting views and difficulties when making decisions about the final intervention. There is no consensus on the size of team to have, but experts think it is important to agree a process for making decisions. In particular, experts recommend that team members understand their roles, rights and responsibilities; document the reasons for decisions made and are prepared to test different options where there are team disagreements.

### Review published research evidence

Reviewing published research evidence before starting to develop an intervention can help to define the health problem and its determinants, understand the context in which the problem exists, clarify who the intervention should be aimed at, identify whether effective or cost-effective interventions already exist for the target population/setting/problem, identify facilitators and barriers to delivering interventions in this context and identify key uncertainties that need to be addressed using primary data collection. Continuing to review evidence throughout the process can help to address uncertainties that arise, for example, if a new substantive intervention component

is proposed then the research evidence about it can be explored. Evidence can change quickly, and keeping up with it by reviewing literature can alert developers to new relevant interventions that have been found to be effective or cost-effective. Developers may be tempted to look for evidence that supports existing ideas and plans, but should also look for, and take into account, evidence that the proposed intervention may not work in the way intended. Undertaking systematic reviews is not always necessary because there may be recent relevant reviews available, nor is it always possible in the context of tight resources available to the development team. However, undertaking some review is important for ensuring that there are no existing interventions that would make the one under development redundant.

### Draw on existing theories
Some developers call their approaches to intervention development 'theory based' when they draw on psychological, sociological, organisational or implementation theories, or frameworks of theories, to inform their intervention.[6] The rationale for drawing on existing theories is that they can help to identify what is important, relevant and feasible to inform the intended goals of the intervention[13] and inform the content and delivery of any intervention. It may be relevant to draw on more than one existing theory. Experts recommend considering which theories are relevant at the start of the development process. However, the use of theories may need to be kept under scrutiny since in practice some developers have found that their selected theory proved difficult to apply during the development process.

### Articulate programme theory
A programme theory describes how a specific intervention is expected to lead to its effects and under what conditions.[14] It shows the causal pathways between the content of the intervention, intermediate outcomes and long-term goals and how these interact with contextual factors. Articulating programme theory at the start of the development process can help to communicate to funding agencies and stakeholders how the intervention will work. Existing theories may inform this programme theory. Logic models can be drawn to communicate different parts of the programme theory such as the causes of a problem, or the mechanisms by which an intervention will achieve outcomes, to both team members and external stakeholders. Figure 1 is an example of a logic model. The programme theory and logic models are not static. They should be tested and refined throughout the development process using primary and secondary data collection and stakeholder input. Indeed, they are advocated for use in process evaluations alongside outcome evaluations in the recent MRC Guidance on process evaluation.[15]

### Undertake primary data collection
Primary data collection, usually involving mixed methods, can be used for a range of purposes throughout the intervention development process. Reviewing the evidence base may identify key uncertainties that primary data collection can then address. Non-participant observation can be used to understand the setting in which the intervention will be used. Qualitative interviews with the target population or patient group can identify what matters most to people, their lived experience or why people behave as they do. 'Verbal protocol', which involves users of an intervention talking aloud about it as they use it,[16] can be undertaken to understand the usability of early versions of the intervention. Pretest and post-test measures may be taken of intermediate outcomes to begin early testing of some aspects of the programme theory, an activity that will continue into the feasibility and evaluation phases of the MRC framework and may lead to changes to the programme theory. Surveys, discrete choice experiments or qualitative interviews can be used to assess the acceptability, values and priorities of those delivering and receiving the intervention.

### Understand the context
Recent guidance on context in population health intervention research identifies a breadth of features including those relating to population and individuals; physical location or geographical setting; social, economic, cultural and political influences and factors affecting implementation, for example, organisation, funding and policy.[17] An important context is the specific setting in which the intervention will used, for example, within a busy emergency department or within people's homes. The rationale for understanding this context, and developing interventions which can operate within it, is to avoid developing interventions that fail during later evaluation because too few people deliver or use them. Context also includes the wider complex health and social care, societal or political systems within which any intervention will operate.[18] Different approaches can be taken to understand context, including reviews of evidence, stakeholder engagement and primary data collection. A challenge of understanding context is that it may change rapidly over the course of the development process.

### Pay attention to future implementation of the intervention in the real world
The end goal of developers or those who fund development is real-world implementation rather than simply the development of an intervention that is shown to be effective or cost-effective in a future evaluation.[7] Many interventions do not lead to change in policy or practice, and it is important that effective interventions inform policy and are eventually used in the real world to improve health and care. To achieve this goal, developers may pay attention early on in the development process to factors that might affect use of the intervention, 'scale up' of the intervention for use nationally or internationally, and sustainability. For example, consideration of the cost of the intervention at an early stage, including as stakeholders official bodies or policymakers that would endorse or accredit the intervention or addressing the challenges of training practitioners in delivering the

intervention may help its future implementation. Implementation-based approaches to intervention development are listed in table 2. Some other approaches listed in this table, such as the Normalisation Process Theory, also emphasise implementation in the real world.

### Design and refine the intervention

The term 'design' is sometimes used interchangeably with the term 'development'. However, it is useful to see design as a specific creative part of the development process where ideas are generated, and decisions are made about the intervention components and how it will be delivered, by whom and where. Design starts with generation of ideas about the content, format, style and delivery of the proposed intervention. The process of design may use creative ways of generating ideas, for example, using games or physically making rough prototypes. Some teams include experts in design or use designers external to the team when undertaking this action. The rationale for a wide-ranging and creative design process is to identify innovative and workable ideas that may not otherwise have been considered.

After generating ideas, a mock up or prototype of the intervention or a key component may be created to allow stakeholders to offer views on it. Once an early version or prototype of the intervention is available, it can be refined (sometimes called optimised) using a series of rapid iterations where each iteration includes an assessment of how acceptable, feasible and engaging the intervention is, leading to cycles of refinements. The programme theory and logic models are important at this point, and developers may test whether some of their proposed mechanisms of action are impacting on intermediate outcomes if statistical power allows. The rationale for spending time on multiple iterations is that problems can be identified and solutions found prior to any expensive future feasibility or evaluation phase. Some experts take a quantitative approach to optimisation of an intervention, specifically the Multiphase Optimization Strategy in table 2, but not all experts agree that this is necessary.

### End the development phase

Seeing this endeavour as a discrete 'intervention development phase' that comes to an end may feel artificial. In practice, there is overlap between some actions taken in the development phase and the feasibility phase of the MRC framework,[1] such as consideration of acceptability and some measurement of change in intermediate outcomes. Developers may return to the intervention development phase if findings from the feasibility phase identify significant problems with the intervention. In many ways, development never stops because developers will continue to learn about the intervention, and refine it, during the later pilot/feasibility, evaluation and implementation phases. The intention may be that some types of intervention continuously evolve during evaluation and implementation, which may reduce the amount of time spent on the development phase. However, developers need to decide when to stop that first intensive development phase, either in terms of abandoning the intervention because pursuing it is likely to be futile or moving on to the next phase of feasibility/piloting testing or full evaluation. They also face the challenge of convincing potential funders of an evaluation that enough development has occurred to risk spending resources on its pilot or evaluation. The decision to end the development phase may be partly informed by practicalities, such as the amount of time and money available, and partly by the concept of data saturation (used in qualitative research) in that the intensive process stops when few refinements are suggested by those delivering or using the intervention during its period of refinement, or these and other stakeholders indicate that the intervention feels appropriate to them.

At the end of the development process, policymakers, developers or service providers external to the original team may want to implement or evaluate the intervention. Describing the intervention, using one of the relevant reporting guidelines such as the Template for Intervention Description and Replication Checklist[19] and producing a manual or document that describes the training as well as content of the intervention can facilitate this. This information can be made available on a website, and, for some digital interventions, the intervention itself can be made available. It is helpful to publish the intervention development process because it allows others to make links in the future between intervention development processes and the subsequent success of interventions and learn about intervention development endeavours. Publishing failed attempts to develop an intervention, as well as those that produce an intervention, may help to reduce research waste. Reporting multiple, iterative and interacting processes in these articles is challenging, particularly in the context of limited word count for some journals. It may be necessary to publish more than one paper to describe the development if multiple lessons have been learnt for future development studies.

### CONCLUSIONS

This guidance on intervention development presents a set of principles and actions for future developers to consider throughout the development process. There is insufficient research evidence to recommend that a particular published approach or set of actions is essential to produce a successful intervention. Some aspects of the guidance may not be relevant to some interventions or contexts, and not all developers are fortunate enough to have a large amount of resource available to them, so a flexible approach to using the guidance is required. The best way to use the guidance is to consider each action by addressing its relevance to a specific intervention in a specific context, both at the start and throughout the development process.

**Acknowledgements** This guidance is based on secondary and primary research. Many thanks to participants in the e-Delphis, consensus conference and qualitative interviews, to members of our Expert Panel and to people who attended workshops discussing this guidance. The researchers leading the update of the MRC guidance on developing and evaluating interventions, due to be published later this year, also offered insightful comments on our guidance to facilitate fit between the two sets of guidance.

**Contributors** AOC and PH led the development of the guidance, wrote the first draft of the article and the full guidance document which it describes, and integrated contributions from the author group into subsequent drafts. All authors contributed to the design and content of the guidance and subsequent drafts of the paper (AOC, PH, LY, LC, NR, KMT, ED, KS). The guidance is based on reviews and primary research. AOC led the review of different approaches to intervention development working with KS. LC led the review of primary research working with KS. PH led the qualitative interview study working with NR, KMT and ED. ED led the consensus exercise working with NR. AOC acts as guarantor.

**Funding** MRC-NIHR Methodology Research Panel (MR/N015339/1). Funders had no influence on the guidance presented here. The authors were fully independent of the funders.

**Competing interests** None declared.

**Patient consent for publication** Not required.

**Provenance and peer review** Not commissioned; externally peer reviewed.

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
