## [Reviewer comments · BMJ Open]

ARTICLE DETAILS

TITLE (PROVISIONAL)	Guidance on how to develop complex interventions to improve health and health care
AUTHORS	O'Cathain, Alicia; Croot, Liz; Duncan, Edward; Rousseau, Nikki; Sworn, Katie; Turner, Katrina; Yardley, Lucy; Hoddinott, Pat

VERSION 1 - REVIEW

REVIEWER	Janet Harris University of Sheffield England
REVIEW RETURNED	26-Mar-2019

GENERAL COMMENTS	This is important guidance addressing an underdeveloped area of intervention development. I think it is a challenge to produce this sort of guidance because the success of intervention development depends upon the approaches used to facilitate the process of collaboration, and collaborative processes are difficult to describe concisely in a single article. If word count allowed, it would be helpful to: (1) have several sentences defining intervention experts and stakeholders - what experience and criterion were used to select them? (2) give examples that illustrate what working closely looks like; (3) discuss when consultation is useful as opposed to co-production; (4) provide examples of creative activities. Under Review, Draw on and Articulate, it could be useful to add phrases that illustrate the group process further e.g. 'Use stakeholder experiences of implementing or participation in interventions to critique existing theories'; 'Use group experiences of delivering or receiving an intervention to articulate programme theories'. In the Development phase, the concept of appropriateness and relevance to context might be useful when deciding when to stop development/ In other words, when stakeholders and early recipients of the intervention indicate that the intervention is appropriate then further development for that particular context can be stopped. I'm not sure what this sentence means: "In practice, some developers may need to fund various parts of the development process opportunistically ..." Are you trying to get at the fact that the development process is dynamic? You mention at the beginning of the paper, but there is something here about the fact that the interventions themselves are dynamic rather than fixed and will often need to be modified over time. Table 2 is very helpful but it needs to be noted that Partnerships can be used for all of the intervention development approaches.
---

	The section after Table 2 provides detailed description of the Framework of Action. Can these sections either be placed together, or can the Framework cross reference to the detailed description? I'd suggest refining the sentences saying that the evidence base for following a particular set of steps is sparse. There are other fields, such as research into design processes, which put forward steps similar to the ones in your guidance...so wouldn't it be more accurate to say that the evidence base is sparse in health research but that your guidance is supported by research in other disciplines?
--	---

REVIEWER	Dr David Maidment School of Sport, Exercise and Health Sciences, Loughborough University, UK
REVIEW RETURNED	29-Mar-2019

GENERAL COMMENTS	The current manuscript builds-on the UK MRC's existing framework for developing and evaluating complex healthcare interventions, providing further detailed guidance (i.e. key principles and actions) that can be applied when undertaking the initial development phase. Overall, the manuscript describes a much-needed novel framework that has the potential to guide intervention developers across multiple disciplines. Having been involved in the development and evaluation of complex healthcare interventions myself, I found this formalised guidance to be very insightful, as well as consistent with my own experiences. In particular, I agree that working with key stakeholders from the outset and employing iterative usability testing are both crucial aspects for the development of complex healthcare interventions. I have the following comments/questions, that the authors could address to improve an already excellent Communication article:  - Page 3, lines 57-60. The rationale for why the authors provide more detailed guidance on intervention development, as opposed to another of three stages outlined in the MRC framework, is not entirely clear. The authors could state explicitly here why detailed guidance on intervention development, which expands on the MRC framework, is needed. - Page 4, lines 12-18. The consensus exercises undertaken by the authors could include greater detail. For example, for the second study, who were the 'key stakeholders'. In addition, how were the 80-items that participants rated generated? I appreciate that the authors may only intend to provide limited detail, as they possibly intend to publish the findings of these studies elsewhere. Could these also be signposted to the reader, if possible? Page 4, lines 26. The authors refer to an 'evidence gap', but it is currently unclear how this was determined. If this originates from the reviews cited in the previous section, the authors could add a reference here also. Page 4, line 52. The authors state here that the development phase is complete when the 'team feels they have an intervention that is
---

	worthwhile evaluating'. Based on the 'End of Development phase' section (page 13, line 29), I am not sure that their earlier assertion is completely accurate. The authors could consider rephrasing this sentence. Page 5, line 32. Much of the information in Table 1 is somewhat repeated in main body of the subsequent text, which I initially found confusing. The authors could condense the Table or amalgamating its contents with the main text. Page 11, line 52. I am not familiar with 'programme theory'. However, it does seem to share a resemblance to the MRC's guidance for the process evaluation of complex interventions (https://www.bmj.com/content/350/bmj.h1258). If appropriate, the authors could consider also incorporating/citing this guidance into their manuscript. Page 12, line 47. The implementation of an intervention in the real-world is inherently complex and poorly understood. A number of papers have been published that attempt to that specify implementation plans/strategies (e.g. https://www.ncbi.nlm.nih.gov/pmc/articles/PMC6194634/). The authors could also include examples of such plans/strategies in this section, which might assist with implantation considerations during intervention development.
--	---

REVIEWER	Marieke J. Schuurmans University Medical Center Utrecht, the Netherlands
REVIEW RETURNED	03-Apr-2019

GENERAL COMMENTS	A growing number of problems in health care can be addressed by so called complex interventions/interventions that have a number of interacting components, that require new behaviours and/or have a variety of outcomes. I fully agree with the authors that the development phase of the MRC framework regarding complex interventions is crucial. Moreover, if the development phase is not thorough, positive outcomes of the intervention are not very hopeful. For this reason we published in 2018 our paper to strengthen the development phase of the MRC framework (Bleijenberg et al, Increasing value and reducing waste by optimizing the development of complex interventions: Enriching the development phase of the Medical Research Council (MRC) Framework, International Journal of Nursing Studies, 79(2018), 86-93). Although the paper of O’Cathain et al, has nice elements, reviewing the paper leaves me with a two major questions. The first, what is the added value of this paper compared to what is already written regarding the development phase, what is new, innovative, distinctive? Richer literature on this subject is published. In addition, crucial elements (such as context) of intervention development are not emphasized enough and the actions / processes described are not sufficiently linked to each other. Second, what is the underlying evidence for this approach, the methods are not clear and many of the finding seem rather coincidental instead of systematic.
--

With regard to the first question:

Table 1 describes the actions that are required and includes processes that are not new but put together in an orderly manner, this also applies to fig. 1 the logic model.

Literature indicates that aspects of evaluation and implementation should actually start at the beginning of the development process (see also STARI guideline, Pinnock, 2017), this is not taken into account.

Table 2 gives a nice overview, but suggests to me as reader to choose one approach in intervention development. Our experience is that multiple approaches can be used simultaneously, for instance user-centered approach, BHC, MRC + van Meijel framework etc. Moreover the complex nature of complex interventions often cannot succeed with one approach, the choice of multiple approaches is important as well as the rationale for choosing them. The authors do mention flexibility.

Context is highlighted in this paper but should get more attention. The latest article by Moore (2019) (regarding system lens approach) is cited, but does not do justice to the content of that paper, system lens approach should certainly have been mentioned and explained, otherwise a different source. (Note: authors refer to Moore, 2018 while it must be 2019).

Reporting guideline: Tidier is cited. But there are several reporting guidelines such as Credec I + II, STARI etc.

Authors hardly mention anything regarding the development of a training during the development phase of an intervention, although this is mentioned as output in the logic model ... training is only mentioned as part of the Tidier (very briefly). While there is a paragraph with the heading: "Pay attention to future implementation of the intervention in the real world", training is a crucial part of that.

With regard to the second question:

- The authors describe three steps: 1) review, 2) interviews, 3) Delphi. It is not clear in this paper what comes from which step. Also details regarding what kind of experts are approached, from which countries, how and what about the qualitative interviews? In addition, I am curious how the item list with 80 items has been compiled, based on what, and what the outcomes are. In the context of transparency.

- Table 1 is called "framework of actions", but is rather a list and not a framework. The table shows "plan the development". Herein the problem analysis is merged with process steps. I don't think that makes sense. Because a good problem analysis is also based on a scientific exercise. In the text below it seems to be described separately.

- Search strategy is missing: how then did the included reviews come to pass? Self-citing another article that is now in press but is not referred to as a review but as a "systematic methods overview". Title: Taxonomy and synthesis of approaches to developing interventions to improve health

- In general, few articles used to substantiate incl. number of reviews.

General: ref 5 is missing, although reference is made to it in the description of table 2

VERSION 1 – AUTHOR RESPONSE

Reviewer: 1 Janet Harris

This is important guidance addressing an underdeveloped area of intervention development. I think it is a challenge to produce this sort of guidance because the success of intervention development depends upon the approaches used to facilitate the process of collaboration, and collaborative processes are difficult to describe concisely in a single article. If word count allowed, it would be helpful to:

(1) have several sentences defining intervention experts and stakeholders - what experience and criterion were used to select them?

We have added the following information about participants in the interview and consensus studies:

The next two phases involved developers and wider stakeholders. Developers were people who had written articles or books detailing different approaches to developing interventions, and people who had developed interventions. Wider stakeholders were people involved in the wider intervention development endeavour in terms of being directors of research funding panels, editors of journals that had published intervention development studies, people who had been public and patient involvement members of studies involving intervention development, and people working in health service implementation.

(2) give examples that illustrate what working closely looks like;

We have added an example:

This could involve a series of workshops and meetings to build relationships over time to facilitate understanding of the problem and generation of ideas for the new intervention. Co-production rather than consultation is likely to be important when buy-in is needed from a set of stakeholders to facilitate the feasibility, acceptability and engagement with the intervention, or the health problem or context involves particularly complex decisions.

(3) discuss when consultation is useful as opposed to co-production;

We have explained when co-production rather than consultation might be needed by inserting the following text after the sentence above:

Co-production involves stakeholders in this decision making whereas with consultation, decisions are made by the research team.

(4) provide examples of creative activities.

We have provided examples of creative activities by inserting the following text:

for example: creative sessions facilitated by a design specialist might involve imagining what versions of the new intervention might look like if designed by various well known global manufacturers. Or creating a patient persona to help people think through the experiences of receiving an intervention.

Under Review, Draw on and Articulate, it could be useful to add phrases that illustrate the group process further e.g. 'Use stakeholder experiences of implementing or participation in interventions to critique existing theories'; 'Use group experiences of delivering or receiving an intervention to articulate programme theories'.

We feel that this would be essential if we were writing only about partnership approaches to intervention development such as co-design or co-production. The guidance is based on consensus from people using a range of approaches so we have not added this language.

In the Development phase, the concept of appropriateness and relevance to context might be useful when deciding when to stop development/ In other words, when stakeholders and early recipients of the intervention indicate that the intervention is appropriate then further development for that particular context can be stopped.

We have added this into the 'End of Development phase' section:

in that the intensive process stops when few refinements are suggested by those delivering or using the intervention during its period of refinement, or these and other stakeholders indicate that the intervention feels appropriate to them.

I'm not sure what this sentence means: "In practice, some developers may need to fund various parts of the development process opportunistically ..." Are you trying to get at the fact that the development process is dynamic? You mention at the beginning of the paper, but there is something here about the fact that the interventions themselves are dynamic rather than fixed and will often need to be modified over time.

We have edited this sentence which now reads:

in practice some developers may not be able to access this funding and may have to fund different parts of the development process from separate pots of money over a number of years.

Table 2 is very helpful but it needs to be noted that Partnerships can be used for all of the intervention development approaches.

We have placed more emphasis on the combining of approaches, a point also made by Reviewer 3, by inserting the following text before table 2:

Many of these approaches share the same actions [4,6] and simply place more emphasis on one or a sub-set of actions. Researchers sometimes combine the use of different approaches in practice to gain the strengths of two or more approaches, as in the 'Combination' category of Table 2.

The section after Table 2 provides detailed description of the Framework of Action. Can these sections either be placed together, or can the Framework cross reference to the detailed description?

Reviewer 2 also had a problem with this. We have added in a sentence in the section 'key actions in intervention development' to alert the reader to the fact that Table 1 offers a summary and further

details are given later in the paper. We explain why we have taken this action in our response to Reviewer 2.

I'd suggest refining the sentences saying that the evidence base for following a particular set of steps is sparse. There are other fields, such as research into design processes, which put forward steps similar to the ones in your guidance...so wouldn't it be more accurate to say that the evidence base is sparse in health research but that your guidance is supported by research in other disciplines?

It is certainly the case that the evidence base to support these actions in health is sparse and we have refined the sentence accordingly. We don't want to go as far as to say that there is evidence from other fields because the way many health researchers look for evidence is through linking processes and outcomes within systematic reviews and this is not necessarily an approach favoured in other fields.

Reviewer: 2 Dr David Maidment

The current manuscript builds-on the UK MRC's existing framework for developing and evaluating complex healthcare interventions, providing further detailed guidance (i.e. key principles and actions) that can be applied when undertaking the initial development phase.

Overall, the manuscript describes a much-needed novel framework that has the potential to guide intervention developers across multiple disciplines. Having been involved in the development and evaluation of complex healthcare interventions myself, I found this formalised guidance to be very insightful, as well as consistent with my own experiences. In particular, I agree that working with key stakeholders from the outset and employing iterative usability testing are both crucial aspects for the development of complex healthcare interventions.

Thank you

I have the following comments/questions, that the authors could address to improve an already excellent Communication article:

- Page 3, lines 57-60. The rationale for why the authors provide more detailed guidance on intervention development, as opposed to another of three stages outlined in the MRC framework, is not entirely clear. The authors could state explicitly here why detailed guidance on intervention development, which expands on the MRC framework, is needed.

We have now explained why this guidance is needed at the end of the introduction section by inserting the following text:

This phase was only briefly outlined in the original MRC guidance and requires extension to offer more help to researchers wanting to develop complex interventions. Bleijenberg and colleagues (2018) brought together learning from a range of guides/published approaches to intervention

development to enrich the MRC framework.[4] There are also multiple sources of guidance to intervention development, embodied in books and journal articles about different approaches to intervention development (for example[5]), and overviews of the different approaches.[6] These approaches and overviews may offer conflicting advice and it is timely to gain consensus on key aspects of intervention development to help researchers to focus on this endeavour. Here we present guidance on intervention development based on a consensus study which we describe below. We present this guidance as an accessible communication article on how to do intervention development, which is aimed at readers who are developers, including those new to the endeavour. We do not present it as a "Research Article" with methods and findings in order to maximise its utility as guidance. Lengthy detail and a long list of references are not provided so that the guidance is focused and user friendly. In addition, the key actions of intervention development are summarised in a single table so that funding panel members and developers can use this as a type of checklist of issues to consider when developing health interventions.

- Page 4, lines 12-18. The consensus exercises undertaken by the authors could include greater detail. For example, for the second study, who were the 'key stakeholders'. In addition, how were the 80-items that participants rated generated? I appreciate that the authors may only intend to provide limited detail, as they possibly intend to publish the findings of these studies elsewhere. Could these also be signposted to the reader, if possible?

We have expanded this section to offer more detail as requested by all the reviewers. We have also added a supplementary document that reports the answers to the Delphi items so that readers can look for items relevant to queries they have about how to do their intervention development. We have inserted the following text:

The next two phases involved developers and wider stakeholders. Developers were people who had written articles or books detailing different approaches to developing interventions, and people who had developed interventions. Wider stakeholders were people involved in the wider intervention development endeavour in terms of being directors of research funding panels, editors of journals that had published intervention development studies, people who had been public and patient involvement members of studies involving intervention development, and people working in health service implementation. We carried out qualitative interviews [7] and then we conducted a consensus exercise consisting of two simultaneous and identical e-Delphi studies distributed to intervention developers and wider stakeholders respectively, and followed this with a consensus workshop. We generated items for the e-Delphi studies based on our earlier reviews and analysis of interview data and asked participants to rate 85 items on a five point scale from 'very' to 'not important' using the question 'when developing complex interventions to improve health, how important is it to'. The distribution of answers to each item is displayed in Supplementary File 1.

Page 4, lines 26. The authors refer to an 'evidence gap', but it is currently unclear how this was determined. If this originates from the reviews cited in the previous section, the authors could add a reference here also.

This does not originate from our reviews. We have rewritten this section and updated it with a very recent reference of a review of reviews showing the lack of research evidence supporting the use of theory in intervention development:

We base this guidance on expert opinion because there is a research evidence gap about which actions are needed in intervention development to produce successful health interventions.

Systematic reviews have been undertaken to determine whether following a specific published approach, or undertaking a specific action, results in effective interventions. Unfortunately this evidence base is sparse in the field of health, largely due to the difficulty of empirically addressing this question.[8,9] Evidence tends to focus on the use of existing theory within intervention development – for example the theory of Diffusion of Innovation, or theories on behaviour change - and a review of reviews shows that interventions developed with existing theory do not result in more effective intervention than those not using existing theory.[10] The authors of this latter review highlight problems with the evidence base rather than dismiss the possibility that existing theory could help produce successful interventions.

Page 4, line 52. The authors state here that the development phase is complete when the '*team feels* they have an intervention that is worthwhile evaluating'. Based on the 'End of Development phase' section (page 13, line 29), I am not sure that their earlier assertion is completely accurate. The authors could consider rephrasing this sentence.

We have rephrased this sentence in line with Table 1:

There will also be iterative cycles of developing a version of the intervention: getting feedback from stakeholders to identify problems, implementing potential solutions, assessing their acceptability, and starting the cycle again until assessment of later iterations of the intervention produces few changes.

Page 5, line 32. Much of the information in Table 1 is somewhat repeated in main body of the subsequent text, which I initially found confusing. The authors could condense the Table or amalgamating its contents with the main text.

Reviewer 1 also found this confusing. We thought about how someone might like to use the Communication Article and thought that researchers might want to print off Table 1 which summarises the key points. We were thinking about busy people looking for shortcuts to knowledge. So we decided to have an expanded Table 1 rather than a very summarised table. We have considered your recommendation and would like to stick with the original table but we have explained our approach to presentation to the reader to reduce the potential for confusing people (see last sentence of the introduction section):

In addition, we summarise the key actions of intervention development in a single table (Table 1) so that funding panel members and developers can use this as a quick reference point for issues to consider when developing health interventions.

If the editors disagree with this, we are happy to reconsider.

Page 11, line 52. I am not familiar with 'programme theory'. However, it does seem to share a resemblance to the MRC's guidance for the process evaluation of complex interventions (<https://www.bmj.com/content/350/bmj.h1258>). If appropriate, the authors could consider also incorporating/citing this guidance into their manuscript.

I am an author on the MRC guidance for process evaluation (Alicia O'Cathain) and you are right that the use of programme theories and logic models is advocated strongly within that guidance. We have added in the Moore et al 2015 reference to our paper because it is helpful to connect readers to other

relevant guidance advocating similar things in different phases of development and evaluation of complex interventions:

Indeed they are advocated for use in process evaluations alongside outcome evaluations in the recent MRC Guidance on process evaluation.[15]

Page 12, line 47. The implementation of an intervention in the real-world is inherently complex and poorly understood. A number of papers have been published that attempt to that specify implementation plans/strategies (e.g. <https://www.ncbi.nlm.nih.gov/pmc/articles/PMC6194634/>). The authors could also include examples of such plans/strategies in this section, which might assist with implantation considerations during intervention development.

We agree that implementation of an intervention in the real world is challenging. The Normalisation Process Theory is the solution proposed in the article you highlight and this is one of the 'theory and evidence based' published approaches to intervention development we show in Table 2. We have added a sentence to the 'future implementation' section to guide the reader to this:

Implementation-based approaches to intervention development are listed in Table 2. Some other approaches listed in this table, such as the Normalisation Process Theory, also emphasise implementation in the real world.

Reviewer: 3 Marieke J. Schuurmans

A growing number of problems in health care can be addressed by so called complex interventions/interventions that have a number of interacting components, that require new behaviours and/or have a variety of outcomes. I fully agree with the authors that the development phase of the MRC framework regarding complex interventions is crucial. Moreover, if the development phase is not thorough, positive outcomes of the intervention are not very hopeful. For this reason we published in 2018 our paper to strengthen the development phase of the MRC framework (Bleijenberg et al, Increasing value and reducing waste by optimizing the development of complex interventions: Enriching the development phase of the Medical Research Council (MRC) Framework, *International Journal of Nursing Studies*, 79(2018), 86-93).

Marieke, your team and our team set out to do a similar piece of work at the same time. It was literally 'great minds think alike'. We applied to the MRC Methodology Research Panel in 2015 to undertake secondary and primary research in preparation for writing guidance for developing interventions to improve health. We started the funded study in 2016. We read your 2018 paper with great interest – it's an excellent paper - and referenced it in our 2019 paper O'Cathain A, Croot L, Sworn K, et al. Taxonomy and synthesis of approaches to developing interventions to improve health: a systematic methods overview. *Pilot and Feasibility Studies* 2019;5:41. In your 2018 paper and our 2019 paper we both took exactly the same approach of synthesising actions taken in a range of published approaches to intervention development. You looked at 8 guides, four of which were nursing and three of which were health promotion/public health, and we looked at 23 guides across a wider range of disciplines. By the time we knew about your paper we already had our review completed and a draft article ready. We published it on the basis of adding value by working across a wider set of guides and disciplines than your paper and acknowledged in our article that we produced similar findings to you.

The paper you have reviewed here is different from your 2018 paper and our 2019 systematic methods overview paper. It builds on our systematic methods overview, a further review, a qualitative interview study with intervention developers and wider stakeholders, and a consensus exercise with international participants, to produce guidance. We set out to write an accessible guide for developers informed by our consensus exercise which in turn was informed by our reviews and interviews. Our guidance goes beyond other guidance embodied in published approaches, and reviews of published approaches. We have clarified this in the Introduction Section:

This phase was only briefly outlined in the original MRC guidance and requires extension to offer more help to researchers wanting to develop complex interventions. Bleijenberg and colleagues (2018) brought together learning from a range of guides/published approaches to intervention development to enrich the MRC framework.[4] There are also multiple sources of guidance to intervention development, embodied in books and journal articles about different approaches to intervention development (for example[5]), and overviews of the different approaches.[6] These approaches and overviews may offer conflicting advice and it is timely to gain consensus on key aspects of intervention development to help researchers to focus on this endeavour. Here we present guidance on intervention development based on a consensus study which we describe below. We present this guidance as an accessible communication article on how to do intervention development, which is aimed at readers who are developers, including those new to the endeavour. We do not present it as a “Research Article” with methods and findings in order to maximise its utility as guidance. Lengthy detail and a long list of references are not provided so that the guidance is focused and user friendly. In addition, the key actions of intervention development are summarised in a single table so that funding panel members and developers can use this as a quick reference point of issues to consider when developing health interventions.

Although the paper of O’Cathain et al, has nice elements, reviewing the paper leaves me with a two major questions. The first, what is the added value of this paper compared to what is already written regarding the development phase, what is new, innovative, distinctive? Richer literature on this subject is published. In addition, crucial elements (such as context) of intervention development are not emphasized enough and the actions / processes described are not sufficiently linked to each other. Second, what is the underlying evidence for this approach, the methods are not clear and many of the finding seem rather coincidental instead of systematic.

The added value of our paper is that it brings together learning from a consensus exercise based on a review of published approaches and qualitative interviews in an accessible format. We address new areas, such as bringing together a team, resources required, and choosing between intervention development approaches. We have deliberately not included lots of detail about the methods, and we have written it as a “Communication Article”, so we can communicate clearly what actions developers may want to consider during the early development phase. We deliberately aimed to be succinct to complement published approaches to intervention development, some of which are highly detailed within books. We based the style of writing on the highly cited MRC guidance on process evaluation by Moore et al 2015 in the BMJ.

‘Understand context’ is an action in our guidance. Different developers and authors of published approaches value different actions. Another reviewer might ask us to make more of stakeholder involvement or the use of theory. We have worked hard to strike a balance across the range of actions and values expressed in the different data sources contributing to our consensus exercise.

With regard to the first question:

Table 1 describes the actions that are required and includes processes that are not new but put together in an orderly manner, this also applies to fig. 1 the logic model.

We agree that few of these actions are new, although some are (see our response to your earlier point). Our intention is to bring together the key issues in an accessible “Communication Article” to provide an overview of actions to help developers. Otherwise they would have to digest an enormous range of literature, some of it offering conflicting advice.

Literature indicates that aspects of evaluation and implementation should actually start at the beginning of the development process (see also STARI guideline, Pinnock, 2017), this is not taken into account.

We totally agree that both of these should occur early in the process. We already say in the key principles section: “Developers may also benefit from looking forward to how the intervention will be evaluated so they can make plans for this, and identify learning and key uncertainties to be addressed in future evaluation”. “Pay attention to future implementation of the intervention in the real world....from the start....” is one of the actions in Table 1.

Table 2 gives a nice overview, but suggests to me as reader to choose one approach in intervention development. Our experience is that multiple approaches can be used simultaneously, for instance user-centered approach, BHC, MRC + van Meijel framework etc. Moreover the complex nature of complex interventions often cannot succeed with one approach, the choice of multiple approaches is important as well as the rationale for choosing them. The authors do mention flexibility.

We agree and we have added a sentence to make this clear and drawn attention to Category 8 ‘Combination’ in Table 2 which describes the situation you rightly say is used by some developers:

Many of these approaches share the same actions [4,6] and simply place more emphasis on one or a sub-set of actions. Researchers sometimes combine the use of different approaches in practice to gain the strengths of two or more approaches, as in the ‘Combination’ category of Table 2.

Context is highlighted in this paper but should get more attention. The latest article by Moore (2019) (regarding system lens approach) is cited, but does not do justice to the content of that paper, system lens approach should certainly have been mentioned and explained, otherwise a different source. (Note: authors refer to Moore, 2018 while it must be 2019).

The guidance is based on consensus and the systems lens did not come up in our reviews or interviews and so was not included in our Delphi exercises. This system lens is rising in popularity for public health interventions. The remit of our study was much wider and included all health and health care complex interventions. We therefore reference it in relation to something that came up strongly in the work we did – the importance of context. Thank you for pointing out that the correct reference is 2019. We had cited the online version of the paper which was published in 2018; and we have now amended this.

Reporting guideline: Tidier is cited. But there are several reporting guidelines such as CredecI I + II, STARI etc.

We have indicated now that TIDIER is one of a number of guidelines rather than the only one. We prefer to do this rather than list a number of them to keep the paper as focused as possible:

Describing the intervention, using one of the relevant reporting guidelines such as TIDieR (Template for Intervention Description and Replication) Checklist, and producing a manual or document that describes the training as well as content of the intervention, can facilitate this.

Authors hardly mention anything regarding the development of a training during the development phase of an intervention, although this is mentioned as output in the logic model ... training is only mentioned as part of the Tidier (very briefly). While there is a paragraph with the heading: "Pay attention to future implementation of the intervention in the real world", training is a crucial part of that.

In the section on 'End the development phase' we propose "producing a manual or document that describes the training as well as content of the intervention"

With regard to the second question:

- The authors describe three steps: 1) review, 2) interviews, 3) Delphi. It is not clear in this paper what comes from which step. Also details regarding what kind of experts are approached, from which countries, how and what about the qualitative interviews? In addition, I am curious how the item list with 80 items has been compiled, based on what, and what the outcomes are. In the context of transparency.

We have added more detail about the methods. This was also requested by the other two reviewers. We have added the answers given to the 85 items in the Delphi exercises in a Supplementary file:

We generated items for the e-Delphi studies based on our earlier reviews and interviews and asked participants to rate 85 items on a five point scale from 'very' to 'not important' using the question 'when developing complex interventions to improve health, how important is it to'. The distribution of answers to each item is displayed in Supplementary File 1.

- Table 1 is called "framework of actions", but is rather a list and not a framework. The table shows "plan the development". Herein the problem analysis is merged with process steps. I don't think that makes sense. Because a good problem analysis is also based on a scientific exercise. In the text below it seems to be described separately.

Our framework is like many frameworks presented in health research e.g. the Theoretical Domains Framework

- Search strategy is missing: how then did the included reviews come to pass? Self-citing another article that is now in press but is not referred to as a review but as a "systematic methods overview". Title: Taxonomy and synthesis of approaches to developing interventions to improve health

This is a "Communication Article" not a review. The systematic methods overview of 23 published approaches to intervention development is now published in a peer reviewed journal (Reference 6).

'Systematic methods overview' is a systematic approach to reviewing research methods described by Gentles SJ, Charles C, Nicholas DB, Ploeg J, McKibbin KA. Reviewing the research methods literature: principles and strategies illustrated by a systematic overview of sampling in qualitative research. Syst Rev. 2016;5:172.

- In general, few articles used to substantiate incl. number of reviews.

We have deliberately kept the reference list short because this is a Communication Article.

General: ref 5 is missing, although reference is made to it in the description of table 2

Thank you for spotting this. We have removed the citation of reference 5 at Table 2 and edited the references for accuracy.

VERSION 2 – REVIEW

REVIEWER	Janet Harris University of Sheffield England
REVIEW RETURNED	29-May-2019

GENERAL COMMENTS	This will be a very useful paper for the process of intervention design. I have comments about stakeholder involvement but it is up to the author team to decide whether to include them. There is a recently published concept analysis that might be useful when explaining the difference between co-production and consultation, which describes involvement on a continuum from periodic and targeted to ongoing and embedded throughout (Hughes M, Duffy C. Public involvement in health and social sciences research: A concept analysis. Health Expect. 2018;21:1183-1190. DOI: 10.1111/hex.12825.) I agree with the emphasis on stakeholder involvement throughout the process. It's well recognised, however, that many researchers need to develop experience in how to involve stakeholders at different stages. Systematic reviews of involvement have noted that stakeholders are typically included in developing tools and materials and under represented during the design, implementation and analysis stages (Shippee et al, 2015). Researchers are challenged to adopt different roles in relation to co-design (Staniszewska & Denegri 2013; Brett 2014). The Cochrane Collaboration has noted the challenges of involving stakeholders in reviewing evidence (see Cochrane ACTIVE Pollock et al Development of the ACTIVE framework to describe stakeholder involvement in systematic reviews. Journal of health services research
---

	& policy. 2019 Apr 18:1355819619841647.) When reviewing published evidence, for example, we found that involving stakeholders was key in identifying essential elements of the intervention that weren't included in previous research (Harris, Croot, Thompson, Springett 2016). Drawing upon existing theories is very important, but it could be quite important to get stakeholders to review their relevance in terms of contextual validity. Noting these issues with involvement, however, may shift the balance of the paper. Perhaps what we need is a pragmatic follow up paper on when and how to include stakeholders.
--	--

REVIEWER	Marieke Schuurmans University Medical Center Utrecht
REVIEW RETURNED	01-Jun-2019

GENERAL COMMENTS	My two concerns regarding the paper have been addressed partially. Most of the issues with regard to the origin of the findings have been clarified. Given the fact that the authors claim this paper is an consensus exercise to enable and guide funding panel members and developers, still, to my opinion more information with regard to the international expert panel (at least expertise - medical, nursing, public health etc, level of expertise and country of origin) should be provided. Also I do not understand why authors did not describe aspects of evaluation and implementation at the beginning of the development process (see also STARI guideline, Pinnock, 2017), as it is known that seeing the whole process of development as separate for evaluation and implementation is a complicating the chances of actual use of any innovation. The same counts for the lack of interest in training (only mentioned as outcome in the logic model), I do not understand why this is not adressed. The acknowledgement that training of professionals - changing practice is no easy - is a neglected area with regard to complex interventions is necessary. Last year we concluded (Smit et al, J Clin Epidemiol, 2018, April, 96, 119-119) based on a systematic analysis of nine complex interventions that not only context, modeling of the processes and outcomes, measurement and reporting of intervention fidelity need more attention but that implementation of effective training for interventionists is needed to enhance the development and replication of future complex interventions. To conclude: although not authors choose not to adress all comments before acceptance I would suggest to add 1) an addendum with regard to the expertise etc of the consensus panel and 2) to spent at least to lines on training, only mentioning this is the logic model/figure 1 is to little.
---

VERSION 2 – AUTHOR RESPONSE

Response to reviewers' comments 26/6/19

Reviewer 1

1. This will be a very useful paper for the process of intervention design. I have comments about stakeholder involvement but it is up to the author team to decide whether to include them.

Thank you. We note that the issues below are optional.

2. There is a recently published concept analysis that might be useful when explaining the difference between co-production and consultation, which describes involvement on a continuum from periodic and targeted to ongoing and embedded throughout (Hughes M, Duffy C. Public involvement in health and social sciences

research: A concept analysis. *Health Expect.* 2018;21:1183-1190. DOI:

10.1111/hex.12825.)

I agree with the emphasis on stakeholder involvement throughout the process. It's well recognised, however, that many researchers need to develop experience in how to involve stakeholders at different stages. Systematic reviews of involvement have noted that stakeholders are typically included in developing tools and materials and under represented during the design, implementation and analysis stages (Shippee et al, 2015). Researchers are challenged to adopt different roles in relation to co-design (Staniszewska & Denegri 2013; Brett 2014). The Cochrane Collaboration has noted the challenges of involving stakeholders in reviewing evidence (see Cochrane ACTIVE Pollock et al Development of the ACTIVE framework to

describe stakeholder involvement in systematic reviews. *Journal of health services research*

& policy. 2019 Apr 18:1355819619841647.) When reviewing published evidence, for example, we found that involving stakeholders was key in identifying essential elements of the intervention that weren't included in previous research (Harris, Croot, Thompson, Springett 2016).

Drawing upon existing theories is very important, but it could be quite important to get stakeholders to review their relevance in terms of contextual validity. Noting these issues with involvement, however, may shift the balance of the paper. Perhaps what we need is a pragmatic follow up paper on when and how to include stakeholders.

We paid a lot of attention to balance when writing the paper. We didn't want to offer more detail, or references, on one aspect of intervention development than another. We agree that the papers you put forward here are excellent but want to maintain the balance of the paper and so have chosen not to reference them. We agree that a follow up paper on when and how to involve stakeholders would be useful. Hughes et al's paper on public involvement could be extended to include the full range of stakeholders and present examples of working with this wider set of stakeholders.

Reviewer 3

1. My two concerns regarding the paper have been addressed partially. Most of the issues with regard to the origin of the findings have been clarified.

We are pleased that we addressed your concerns.

2. Given the fact that the authors claim this paper is a consensus exercise to enable and guide funding panel members and developers, still, to my opinion more information with regard to the international expert panel (at least expertise -medical, nursing, public health etc, level of expertise and country of origin) should be provided.

We have included details in a new Appendix 2.

3. Also I do not understand why authors did not describe aspects of evaluation and implementation at the beginning of the development process (see also STARI guideline, Pinnock, 2017), as it is known that seeing the whole process of development as separate for evaluation and implementation is a complicating the chances of actual use of any innovation.

We understand that this is not a compulsory point to address but wanted to explain that we have not ignored the comment. We have reread STARI and still do not think that reporting standards for implementation studies are core to our paper. We already include paying attention to evaluation and implementation during the intervention development process. One of the key actions we present in our paper is paying attention to future implementation in the real world. This is highlighted in the abstract. One of the key principles we present is looking forward to future evaluation.

4. The same counts for the lack of interest in training (only mentioned as outcome in the logic model), I do not understand why this is not addressed. The acknowledgement that training of professionals - changing practice is no easy - is a neglected area with regard to complex interventions is necessary. Last year we concluded (Smit et al, J Clin Epidemiol, 2018, April, 96, 119-119) based on a systematic analysis of nine complex interventions that not only context, modeling of the processes and outcomes, measurement and reporting of intervention fidelity need more attention but that implementation of effective training for interventionists is needed to enhance the development and replication of future complex interventions.

We have added this challenge around training into the section on 'pay attention to future implementation of the intervention in the real world'.

5. To conclude: although authors choose not to address all comments before acceptance I would suggest to add 1) an addendum with regard to the expertise etc of the consensus panel and 2) to spend at least a few lines on training, only mentioning this in the logic model/figure 1 is a little.

We have included the requested addendum and addition of the point about training.